



# Spatial and temporal stable water isotope data from the upper snowpack at the EastGRIP camp site, NE Greenland sampled in summer 2018

Alexandra M. Zuhr[1], Sonja Wahl[2], Hans Christian Steen-Larsen[3], Maria Hörhold[4], Hanno Meyer[1], Vasileios Gkinis[5], and Thomas Laepple[1,6]

[1]Alfred-Wegener-Institut Helmholtz Zentrum für Polar- und Meeresforschung, Research Unit Potsdam, Potsdam, Germany
[2]School of Architecture, Civil and Environmental Engineering, Ecole Polytechnique Fédérale de Lausanne, Lausanne, Switzerland
[3]University of Bergen and Bjerknes Centrefor Climate Research, Bergen, Norway
[4]Alfred-Wegener-Institut Helmholtz Zentrum für Polar- und Meeresforschung, Research Unit Bremerhaven, Bremerhaven, Germany
[5]Niels Bohr Institute, Physics of Ice, Climate and Earth, Copenhagen, Denmark
[6]MARUM – Center for Marine Environmental Sciences and Faculty of Geosciences, University of Bremen, Bremen, Germany

**Correspondence:** Alexandra M. Zuhr (alexandra.zuhr@awi.de)

**Abstract.**

Stable water isotopes stored in snow, firn and ice are used to reconstruct environmental parameters. The imprint of these parameters at the snow surface as well as their preservation in the upper snowpack is determined by a number of processes influencing the recording of the environmental signal.

Here, we present a dataset of approximately 3800 snow samples analysed for their stable water isotope composition which were obtained during the summer season at the deep drilling site of the East Greenland Ice Core Project in northeast Greenland. Sampling was carried out every third day between 14 May and 3 August 2018 along a 39 m long transect. Three depth intervals in the top 10 cm were sampled on 30 positions with a higher resolution closer to the surface (0 - 1 cm and 1 - 4 cm depth vs. 4 - 10 cm). The sample analysis was carried out at two renowned stable water isotope laboratories and produced isotope data with an overall highest uncertainty of 0.09 ‰ for $\delta^{18}$O and 0.8 ‰ for $\delta$D.

     This unique dataset shows strongest $\delta^{18}$O variability closest to the surface, damped and delayed variations in the lowest layer and a trend towards increasing homogeneity towards the end of the season, especially in the deepest layer. Additional information on the snow height and its temporal changes suggests a non-uniform spatial imprint of the seasonal climatic information in this area potentially following the stratigraphic noise of the surface.

The data can be used to study the relation between snow height (changes) as well as the imprint and preservation of the isotopic composition at a site with 10 - 14 cm w.eq. yr⁻¹ accumulation. The high temporal resolution sampling allows additional analyses on (post-)depositional processes, such as vapour-snow exchange. The data can be accessed at https://doi.pangaea.de/10.1594/PANGAEA.956626 (Zuhr et al., 2023).



# 1 Introduction

Stable water isotopes measured in ice cores are widely used as proxies for past temperatures (e.g., Dansgaard, 1964; Jouzel et al., 2003; Brook and Buizert, 2018). For reliable reconstructions, it is essential to understand the processes occurring during the signal formation and imprint as well as the modifications of the signal during preservation.

Snowfall above ice sheets contains a signature of the atmospheric temperature. Snowfall itself can be intermittent in space and time (e.g., Persson et al., 2011) and might after the initial deposition be affected by wind erosion leading to mass redis-

tribution to other positions (e.g., Li and Pomeroy, 1997a, b; Filhol and Sturm, 2019). Snow erosion and redeposition are not spatially homogeneous but influenced by surface features, such as dunes and sastrugi (e.g., Fisher et al., 1985; Picard et al., 2019; Zuhr et al., 2021). Hence, accumulation is characterised by the seasonality of snowfall, the meteorological conditions, e.g., wind speed and direction, as well as the stratigraphic features at a specific site.

Accumulation intermittency influences the recording of the environmental information and introduces a significant amount

of noise to the proxy time series (e.g., Casado et al., 2018, 2020). An additional signal imprint occurs during and after the deposition when the stable water isotopes experience exchange processes with the atmosphere and the snow beneath (e.g., Steen-Larsen et al., 2014; Dadic et al., 2015; Ritter et al., 2016; Wahl et al., 2022). Moreover, isotopic diffusion within the snow and firn column smoothes the original signature (Johnsen et al., 2000).

Comparison of isotopic records sampled on snow trenches revealed a large spatial variability of the snowpack and firn

column, which causes a low signal-to-noise ratio of the proxy data, i.e., stable water isotopes, and associated stratigraphic noise in regions of low accumulation (e.g., Münch et al., 2016, 2017). Repeatedly sampled snow profiles with additional information on the snow height at each sampling location illustrated the buildup of this stratigraphic noise and the associated noise in the isotope data (Zuhr et al., 2023). However, questions still exist on the signal contribution from precipitation, vapour-snow exchange, and other processes relevant for the isotope ice core signal.

In recent years, (modeling and observational) studies consider a number of processes, such as precipitation intermittency and diffusion (Casado et al., 2020), vapour-snow exchange (e.g., Touzeau et al., 2018; Hughes et al., 2021; Wahl et al., 2022) or snow redistribution (Libois et al., 2014). These and many other studies contribute to pinning down the processes which are not fully understood yet and which need a better quantification. Thus, the motivation for this study is to answer questions regarding (post-)depositional modifications of the isotopic composition in surface snow, the representativeness of individual

snow profiles and ultimately the preserved signal in the upper snowpack, firn and ice in the end. To answer these questions, the following dataset is a case study that provides insights into the spatial and temporal variability of stable water isotopes in the upper snowpack.



## 2 Description of the dataset

### 2.1 Study area

Snow sampling was performed in a clean snow area next to the campsite of the East Greenland Ice-Core Project (EastGRIP) in the accumulation zone of the Greenland Ice Sheet (75° 38' N, 36° W; 2708 m a.s.l.; Fig. 1a) (Dahl-Jensen et al., 2019). An automatic weather station (AWS) was installed in the vicinity of the study site (Fig. 1b) in 2016 by the Programme for Monitoring of the Greenland Ice Sheet (PROMICE) and provides continuous meteorological data (Fausto and van As, 2019). The study area is characterised by an annual temperature of -26.5 °C (daily averages between -61.6 and -1.9 °C) and average

wind speed of 5.5 m s$^{-1}$ (daily averages between 1.6 and 11.2 m s$^{-1}$) mainly from WSW (240°) for the year 2018 based on data from the PROMICE AWS. The mean accumulation rate ranges around 10 - 14 cm w.eq. yr$^{-1}$ (Schaller et al., 2016; Karlsson et al., 2020).

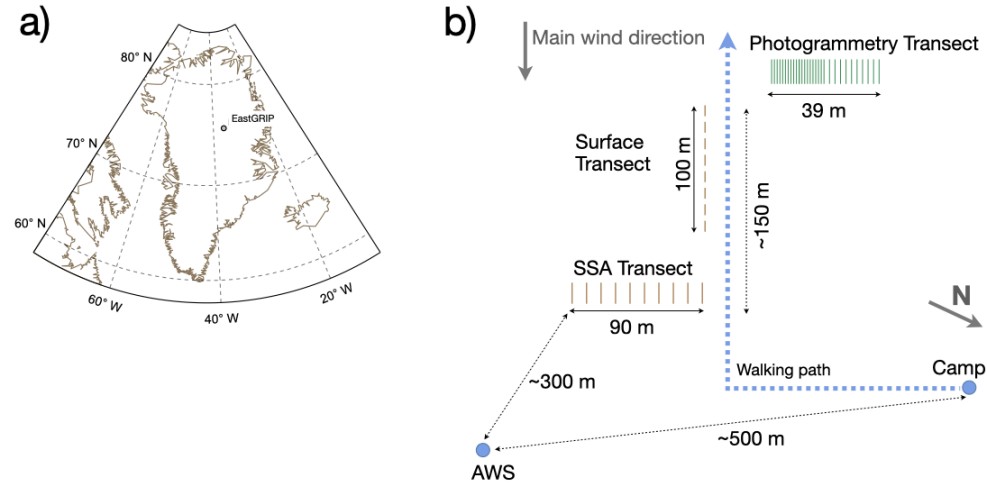

**Figure 1.** The location of the study site next to the EastGRIP camp site is illustrated in a). The organisation of the study site is shown in b) including the snow height measurement and the snow sampling. Specific surface area (SSA) and surface transect are used in complementary studies and mentioned in section 2.5.

### 2.2 Snow sampling

Snow sampling was carried out at 30 positions along a 39 m long transect (Fig. 2) every three days between 14 May and 3

August 2018. The sampling positions had a spacing of 1 m for the first 20 positions and 2 m for the remaining 10 positions (Fig. 2a). The individual positions were marked with green glass fibre sticks which were also used for the additional photogrammetry study performed at the same site during the same observation period (Fig. 2c). Each sampling position was additionally marked



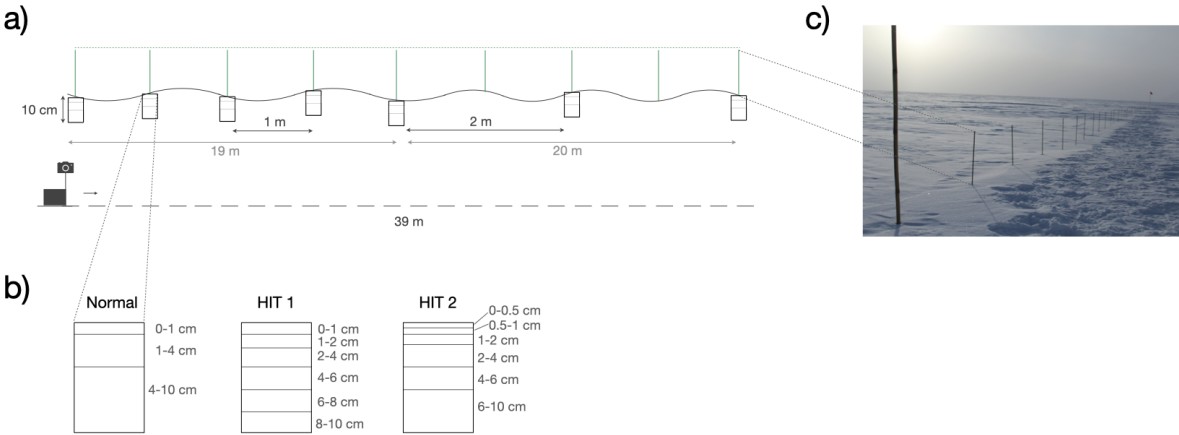

**Figure 2.** Sampling setup showing the snow sampling as well as the snow height measurement is illustrated in (a. Snow was sampled for three depth layers (0 - 1 cm, 1 - 4 cm and 4 - 10 cm) during the normal mode as shown in b). During the high intensity sampling modes (HIT 1 and HIT 2) six depth layers were sampled as indicated and detailed in Table 1. Glass fibre sticks as shown in a) and c) were used for indications of the sampling positions as well as for georeferencing of the photogrammetry. The dashed horizontal line indicates that all glass fibre sticks were levelled to the same height.

with a tiny wooden stick. To avoid re-sampling of the exact same snow and redistribution caused by the sampling itself, the sampling position moved each time some centimetres towards the wind direction.

Three samples were taken at each position (0 - 1 cm, 1 - 4 cm and 4 - 10 cm, Fig. 2b), airtightly stored in Whirl-Paks and transported to Germany in a frozen state. During two periods, sampling was performed every day for seven consecutive days with six sampled depth layers and is referred to as high intensity sampling (HIT). These periods were chosen based on the meteorological conditions during the acquisitions. The first HIT period followed a major snowfall event to study the signal imprint from the surface penetrating into the upper snowpack. This period (HIT 1) was from 8 to 14 June 2018. During that

period samples were taken only at ten sampling positions due to time constraints. The sampled depth intervals were 0 - 1 cm, 1 - 2 cm, 2 - 4 cm, 4 - 6 cm, 6 - 8 cm and 8 - 10 cm (Fig. 2b) at the positions 1, 6, 11, 16, 20, 22, 24, 26, 28 and 30. The second high-resolution sampling period (HIT 2) was from 24 to 30 July 2018 covering a snowfall-free period to focus on vapour-snow exchange processes. 25 positions were sampled (positions 1 - 10, 12, 14, 16, 18, 20 - 30) with the depth intervals 0 - 0.5 cm, 0.5 - 1 cm, 1 - 2 cm, 2 - 4 cm, 4 - 6 cm and 6 - 10 cm. The vertical resolution was changed in order to obtain a higher sampling

resolution closer to the surface where most changes are expected due to (post-)depositional processes (Hughes et al., 2021; Wahl et al., 2022). Respective days with the different snow sampling schemes are also listed in Table 1.





**Table 1.** Detailed information on the sampling days (Date), the respective mode (Mode; n = normal, HIT 1 = high intensity sampling scheme 1, HIT 2 = high intensity sampling scheme 2; explained in the text and Fig. 2b) for each sampling, the number of samples (#), the laboratory of the measurements (Lab; AWI-P = AWI Potsdam, PICE = Physics of Ice, Climate and Earth, University of Copenhagen, Denmark) as well as the availability of a digital elevation model (DEM) from Zuhr et al. (2021).

| Date | Mode | # | Lab | DEM | Date | Mode | # | Lab | DEM | Date | Mode | # | Lab | DEM |
|------|------|---|-----|-----|------|------|---|-----|-----|------|------|---|-----|-----|
| 14.05. | n | 90 | PICE | | 12.06. | HIT 1 | 60 | AWI-P | x | 17.07. | n | 90 | PICE | x |
| 17.05. | n | 90 | AWI-P | | 13.06. | HIT 1 | 60 | AWI-P | | 20.07. | n | 90 | AWI-P | x |
| 20.05. | n | 90 | AWI-P | x | 14.06. | HIT 1 | 180 | AWI-P | x | 23.07. | n | 90 | P-ICE | x |
| 23.05. | n | 90 | AWI-P | x | 17.06. | n | 90 | PICE | | 24.07. | HIT 2 | 150 | AWI-P | x |
| 26.05. | n | 90 | AWI-P | x | 20.06. | n | 90 | PICE | x | 25.07. | HIT 2 | 150 | AWI-P | x |
| 29.05. | n | 90 | AWI-P | x | 23.06. | n | 90 | AWI-P/PICE | | 26.07. | HIT 2 | 150 | AWI-P | x |
| 01.06. | n | 90 | AWI-P | x | 26.06. | n | 90 | AWI-P/PICE | | 27.07. | HIT 2 | 150 | AWI-P | x |
| 05.06. | n | 90 | PICE | x | 29.06. | n | 90 | PICE | | 28.07. | HIT 2 | 150 | AWI-P | |
| 07.06. | n | 90 | AWI-P | | 02.07. | n | 90 | PICE | | 29.07. | HIT 2 | 150 | AWI-P | |
| 08.06. | HIT 1 | 60 | PICE | x | 05.07. | n | 90 | AWI-P | | 30.07. | HIT 2 | 150 | AWI-P | |
| 09.06. | HIT 1 | 60 | PICE | | 08.07. | n | 90 | PICE | | 03.08. | n | 90 | AWI-P | |
| 10.06. | HIT 1 | 60 | PICE | | 11.07. | n | 90 | AWI-P | x | | | | | |
| 11.06. | HIT 1 | 180 | AWI-P | x | 15.07. | n | 90 | AWI-P | | | | | | |

## 2.3 Stable water isotope measurements

Stable water isotope measurements were performed in two laboratories. For most sampling days, all samples were measured in one laboratory. Only two sampling days were split and measured in both laboratories (i.e. 23.06. and 26.06.2018). Table 1
indicates for each sampling day in which laboratory the measurement was performed.

About 70 % of the stable water isotope measurements were performed in the ISOLAB Stable Isotope Facility at the Alfred Wegener Institute (AWI) in Potsdam, Germany, using a Picarro Inc. cavity ring-down spectrometer (model L2140-*i*) with high precision vaporizer (A0211) and autosampler (A0325). The measurement protocol was adjusted to the expected range of isotope values. Hence, the number of injections for samples and reference waters was three, except for the initialisation block
in the beginning of the measurement sequence with standards which were injected six times. All data was calibrated to the VSMOW-SLAP scale. A post-run correction, including memory and drift correction as well as normalisation and calibration, was performed following van Geldern and Barth (2012) using the calibration algorithm described in Münch et al. (2016). The measurement uncertainty of this specific analysis is derived from an independent control reference water which was measured during each run but not used for the calibration. The root mean square deviation of the difference between the expected and the
measured values is used as a measure of uncertainty and is 0.09 ‰ for $\delta^{18}$O and 0.8 ‰ for $\delta$D.

The remaining ~30 % of the samples were measured in the Stable Isotope Laboratory of the Institute for Physics of Ice, Climate and Earth (PICE), Niels Bohr Institute at the University of Copenhagen in Copenhagen, Denmark. A cavity ring-down spectrometer from Picarro Inc. (model L2140-*i*) was used as well but using a high throughput, low volume vaporiser (Picarro-



A0212 – discontinued model as of 2016). The initialisation block in the beginning of the sequences was injected 20 times
per reference water. The number of injections for all successive samples was four. A detailed list of the injection protocol is
provided in Table 3 in Gkinis et al. (2021). No memory correction was applied since the high throughput vaporiser reduced the
amount of memory in the cavity to a level that no correction is necessary for a 12-hour measurement run (Gkinis et al., 2021).
The data were calibrated on the VSMOW-SLAP scale. These measurement runs also contained an independent reference water
which was used to estimate the uncertainty which is 0.04 ‰ for $\delta^{18}$O and 0.33 ‰ for $\delta$D.

All isotopic ratios are reported in ‰ following the delta-notation

$$\delta = \left( \frac{R_S sample}{R_{reference}} \right) - 1 \tag{1}$$

(Craig, 1961) with $R_{sample}$ as the isotopic ratio of the sample and $R_{reference}$ the ratio of an in-house reference water which is
calibrated against the international VSMOW-SLAP scale. The second-order parameter deuterium-excess was calculated from
the data following

$$\text{d-excess} = \delta D - 8 \cdot \delta^{18}O. \tag{2}$$

Cross measurements between both laboratories were performed by measuring the same samples and reference waters in both
laboratories using their respective methods. The samples were kept frozen until the distribution into glass vials at the laboratory
at AWI in Potsdam, Germany. The vials were not filled completely but had a small headspace, as is the usual practice in this
laboratory. The vials were kept cold during the transport to PICE in Copenhagen, Denmark, to avoid any exchange between the
headspace and the water. The raw data were calibrated in each laboratory following their usual procedure as described above.
The comparison resulted in a root mean square deviation of 1.45 ‰ for $\delta^{18}$O and 1 ‰ for $\delta$D.

### 2.4 Meteorological conditions during the 2018 summer season

The observation period between 14 May and 3 August 2018 was characterised by a mean temperature of -14.3 °C (hourly
averages between -30.6 and +0.3 °C; Fig. 3). The average wind speed was 4.1 m s$^{-1}$ (hourly maximum values up to 11.7 m s$^{-1}$)
from a west-southwest direction (236° ± 42°). The latent heat flux shows a diurnal cycle with a maximum during the day and
a minimum during night. Strong fluxes are observed for instance between 4 and 9 June and are accompanied by an increase in
temperature and specific humidity.

Conditions favourable for snow drift events, i.e. 100-hour average above 4 m s$^{-1}$ (Groot Zwaaftink et al., 2013), were present
for 49 % of the time. Snowfall was manually documented on 29 days during the 80 days-long observation period (Fig. 3). No
manual documentation of snowdrift events was performed.

### 2.5 Additional data from this study site

Recent studies performed at the EastGRIP site covered vapour-fluxes and vapour-snow exchange processes (Hughes et al.,
2021; Wahl et al., 2021, 2022) and snow metamorphism (Harris Stuart et al., 2023). A photogrammetry Structure-from-Motion
approach was performed in the same study area along the same transect during the same season in 2018. This dataset provides





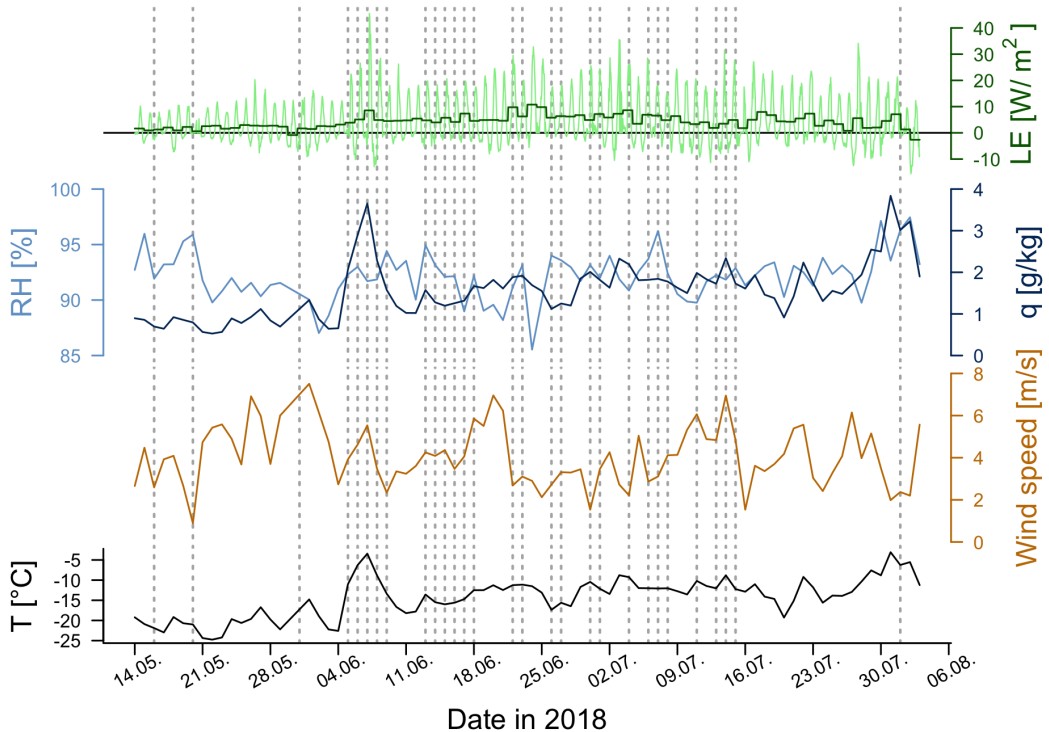

**Figure 3.** Meteorological conditions during the observation period from 14 May to 3 August 2018 from the PROMICE weather station (Fausto and van As, 2019). The 10-minute high-resolution data is averaged to daily values for relative humidity with respect to ice (RH), specific humidity (q), wind speed and temperature (T). The data showing latent heat flux (LE) is averaged to hourly (light green) and daily values (i.e. net LE, dark green). Vertical grey lines show manually documented snowfall events (n = 29) throughout the observation period.

125    high-resolution near-daily digital elevation models (DEMs) with a horizontal resolution of 1 cm and a sufficient accuracy for the purpose of this study (RMSE of 1.3 cm (Zuhr et al., 2021)). It is used to characterise spatiotemporal patterns of snow accumulation and erosion and offers insights into changes of surface structures. A detailed analysis of this surface elevation dataset is presented in Zuhr et al. (2021). The surface roughness decreased during the observation period with an overall flattening of the snow surface. Surface features, such as dunes and sastrugi were more pronounced in May showing a flattening

130    of the surface and a reduction/decrease in surface roughness towards August 2018.

Additional datasets for the summer season in 2018 are available covering stable water isotope data from two surface transects (Fig. 1b; SSA Transect: Steen-Larsen et al. (2022a); consolidate samples from the Surface Transect for the depth intervals 0 - 0.5 cm, 0 - 1 cm and 0 - 2 cm: Hörhold et al. (2022)) as well as from continuous measurements of the isotopic composition in the water vapour (Steen-Larsen and Wahl, 2022).

135    A combination of photogrammetry Structure-from-Motion and stable water isotope data was used in Zuhr et al. (2023) to study the buildup of the isotopic signal in the upper snowpack during the summer season 2019 at the EastGRIP camp site. Their





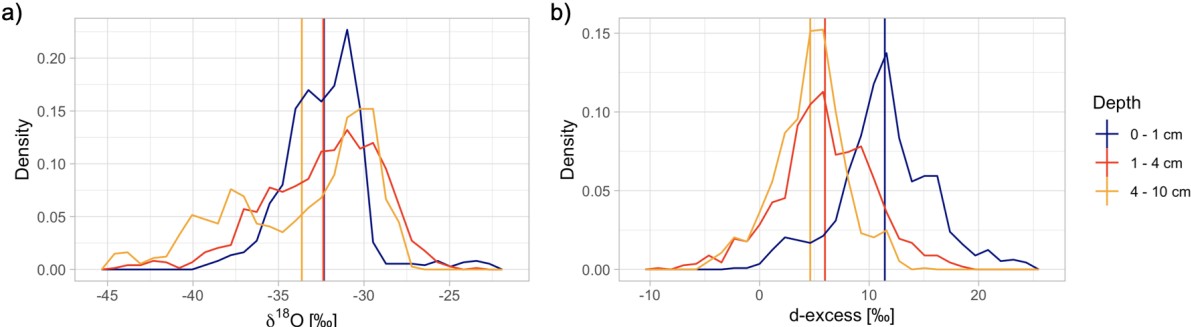

**Figure 4.** Distribution of individual samples $\delta^{18}$O and d-excess values from the normal sampling mode as well as averages from the high resolution samplings per depth layer over the whole season. The vertical lines indicate the respective mean for each sampled depth layer.

study is complementary to the dataset presented here because their setup has a higher vertical and lower temporal resolution over a greater depth (30 cm).

## 3 Results

### 3.1 Description of the data

The dataset consists of 3777 individual snow samples and, hence, isotope data points, which were sampled on 37 days between 14 May and 3 August 2018 (Table 1). The $\delta^{18}$O data have a right skewed distribution and spans a range from -44.6‰ to -22.7‰ with a mean of -32.6‰ and a standard deviation of 3.4‰ (Fig. 4a). The uppermost sampled layer (0 - 1 cm) has the highest $\delta^{18}$O values while the deepest sampled layer (4 - 10 cm) contains the lowest values on average with an absolute minimum value of -44.6‰ (Fig. 4a). The standard deviation across all samples for one depth layer increases from 2.3‰ for the uppermost layer via 3.4‰ for the layer from 1 - 4 cm to 4.2‰ for the lowest layer. The range of individual $\delta^{18}$O values is largest for the interval 1 - 4 cm with 21‰ compared to 16.8‰ for 0 - 1 cm and 17.3‰ for 4 - 10 cm.

Individual values for the second-order parameter d-excess range from -9‰ to 24.6‰ (Fig. 4b). Values for the sampled depth layer are highest for the surface layer with an overall mean of 11.4‰ (ranges between -2.3‰ and 24.6‰). The following layers have mean values of 6‰ (ranges between -9‰ and 18.8‰) and 4.6‰ (ranges between -5‰ and 14.5‰) for the lowest layer.

### 3.2 Spatial and temporal variability of $\delta^{18}$O

The evolution of the isotopic composition ($\delta^{18}$O, Figs. 5 and 6) is characterised by a large increase in surface and subsurface values around 5 June after a snowfall event (Fig. 3), which also affected the lowest sampled layer. Around 14 July, a drop in $\delta^{18}$O in the surface and subsurface layers down to 4 cm depth was observed. The middle layer (1 - 4 cm) shows the highest $\delta^{18}$O composition in the period between 10 June and 11 July while the surface layer (0 - 1 cm) has on average lower values. The second and the third depth layer, i.e. 1 - 4 cm and 4 - 10 cm, show a large spatial variability in May with a decreasing trend



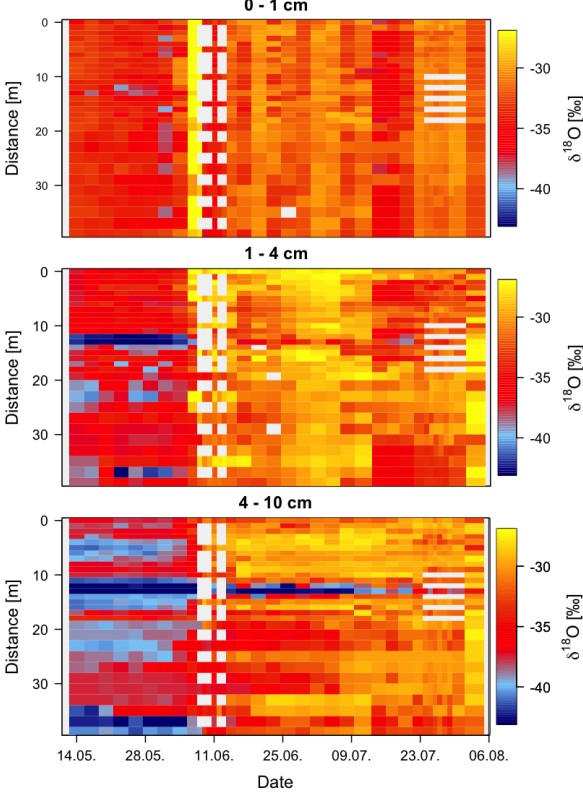

**Figure 5.** Spatial and temporal view of the $\delta^{18}$O data for all sampling days. The date is given in DD.MM. of 2018. The high-resolution sampling periods from 8 to 14 June and 24 to 39 July (Table 1) are visible by variations in the spatial sample coverage (white areas) and are averaged to the depth layers of the normal sampling scheme.

towards the end of the observation period (Fig. 5). These low values persist until mid July at the positions between 10 - 14 m, possibly due to the underlying topography.

160     The surface layer is characterised by faster variations while the layers beneath follow slower and the variations are damped (Fig. 6). The strong increase in $\delta^{18}$O values for the surface layer between 1 and 7 June is recorded in the deeper layer with a damped magnitude while the following drop is not represented in the deepest layer at all. This period was also characterised by snowfall and warm temperatures (Fig. 3).

### 3.3 Spatial and temporal variability of d-excess

The individual depth layers have similar variability in d-excess with standard deviation of 4.4 ‰, 4.3 ‰ and 3.2 ‰ from the top
165  to the bottom layer (Fig. 7). The highest d-excess values are observed in the top layer at 26 June 2018 coinciding with snowfall events (Fig. 3) while the layer from 1 - 4 cm is characterised by lower and partly negative values at the same time. After this event, d-excess in the top layer decreases with time but remains higher than in the layers beneath.

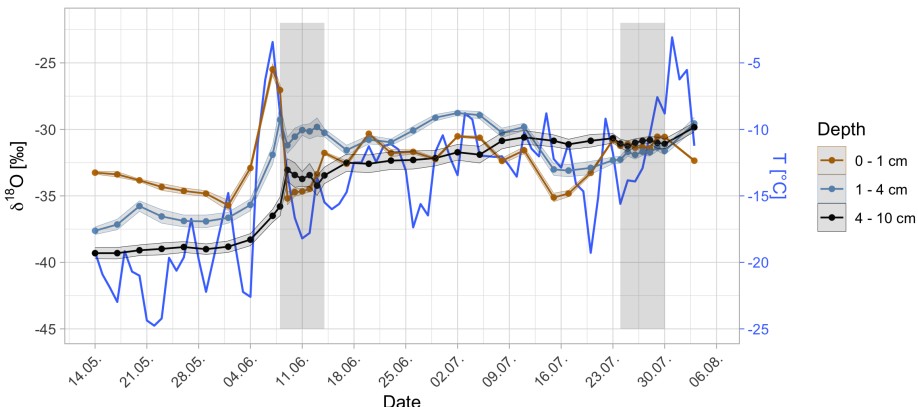

**Figure 6.** Temporal evolution of the spatially averaged $\delta^{18}$O information and the daily temperature (blue line, from the nearby AWS). Each data point is an average across 30 samples along the 39 m long transect. The shading shows the standard error. High-resolution sampling periods are averaged to the overall sampling resolution and are indicated with grey bars.

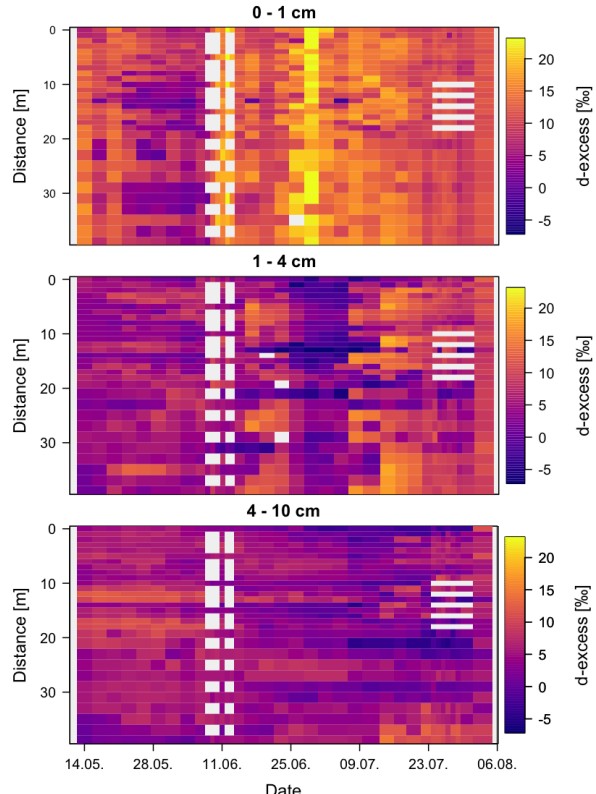

**Figure 7.** Spatial and temporal view of the d-excess data for all sampling days. The date is given in DD.MM. of 2018. The high-resolution sampling periods from 8 to 14 June and 24 to 39 July (Table 1) are visible by white spots due to a reduced spatial sampling coverage.

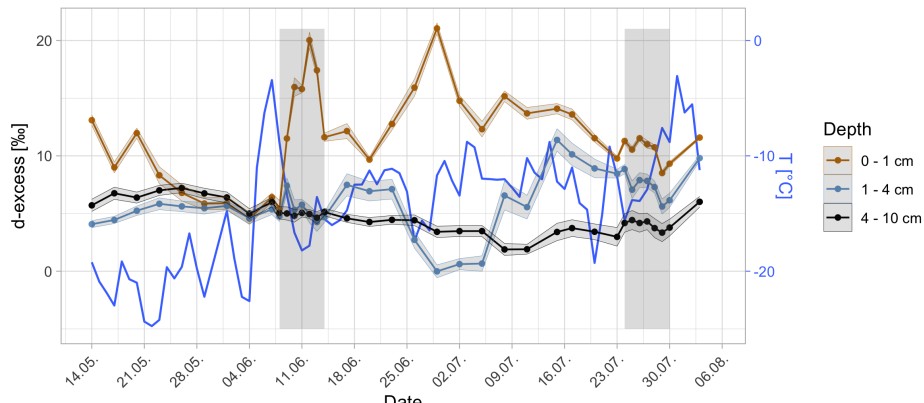

**Figure 8.** Temporal evolution of the spatially averaged d-excess and the daily temperature (blue line, from the nearby AWS). Each data point is an average across 30 samples along the 39 m long transect. The shading shows the standard error. HIT periods are averaged to the overall sampling resolution of three depth layers and indicated with grey bars.

The layer from 4 - 10 cm depth has the lowest variability spatially and also temporally (Fig. 8) with only slight changes between 1.9 and 7.2 ‰ throughout the observation period. The top layer from 0 - 1 cm shows the strongest variability with averaged values above 20 ‰. The lowest d-excess values around 0 ‰ are observed in the layer from 1 - 4 cm during the end of June and beginning of July which seem to coincide with a large spike in the surface layers d-excess.

### 3.4 High-resolution sampling periods

The first high-resolution sampling period (HIT) started on 8 June after snowfall events during the previous days (Fig. 3) with comparably warm temperatures. Snowfall-free days with low wind speeds (Fig. 3) followed afterwards. $\delta^{18}$O shows a drop in the surface layer (0 - 1 cm) and a smaller drop in the layer form 1 - 2 cm while the layers beneath show less variability (Fig. 9). Similarly, the surface layer shows the largest changes in d-excess (5.4 to 20 ‰) while the values in the layers from 1 - 10 cm remain almost constant (ranging between 2.9 and 9.2 ‰).

The isotope composition during the second HIT period (Fig. 10) is characterised by small variations in $\delta^{18}$O with time and slight differences between the individual layers. The overall range of averaged $\delta^{18}$O per depth layer is small with values between -32.6 and -30.3 ‰. The d-excess values, however, show a larger spread across the sampled layers (between 2.1 to 14.2 ‰).

### 4 Discussion

This study presents $\delta^{18}O$ and d-excess from the surface down to 10 cm depth at the EastGRIP camp site. These data can be used to study the evolution of the isotopic composition in the surface snow and upper snowpack to improve the understanding of the overall proxy signal formation and preservation stored in the isotopic composition. The data can further be used to



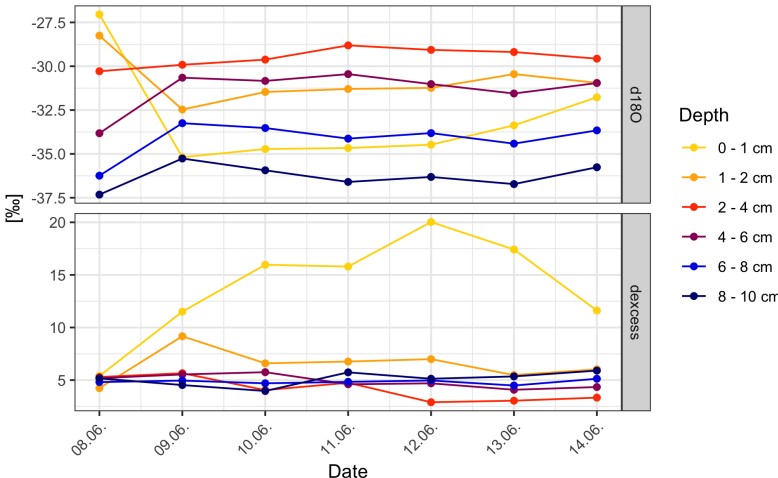

**Figure 9.** $\delta^{18}$O and d-excess values during the first HIT period from 8 to 14 June 2018. The colour code indicates the sampled depth layer. On 11 and 14 June, samples were taken at all 30 positions while on the other days, only at 10 positions.

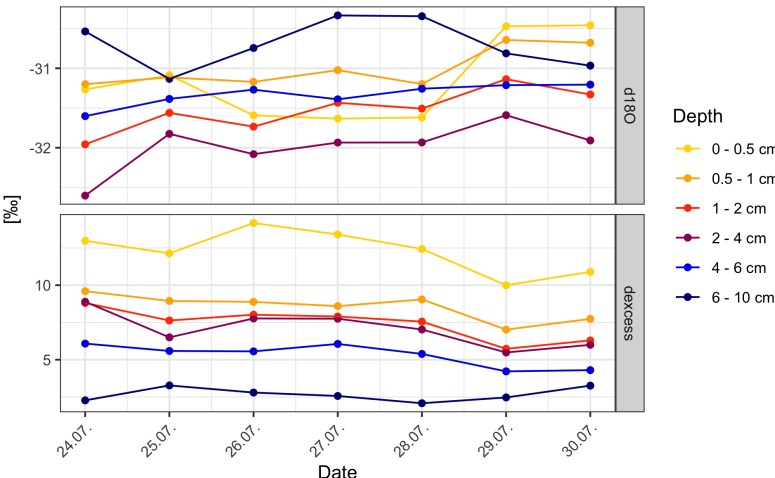

**Figure 10.** $\delta^{18}$O and d-excess values during the first HIT period from 24 to 30 July 2018. The colour code indicates the sampled depth layer. Samples were taken at 25 positions in total, the first 10 with 1 m spacing, followed by 2 m spacing for the remaining 15 positions.

investigate fluxes between the snow surface and the atmosphere as well as within the snowpack, e.g. via the implementation of fluxes and isotopes in models, such as MAR (e.g., Dietrich et al., 2023) and ECHAM-wiso (Cauquoin and Werner, 2021) and comparing these to these data. The outstanding characteristic of the presented dataset is the temporal sampling resolution, the spatial coverage which averages out features from single locations, and the high vertical resolution.



## 4.1 Limitations of the dataset

The maximum sampled depth of 10 cm restricts the temporal coverage of the stable water isotope dataset to some months, depending on the accumulation conditions at each specific sampling site along the 39 m long transect. Hence, the data does not cover an entire seasonal cycle impeding interpretations on a seasonal or annual scale. Moreover, with new snow accumulating over time, we cannot trace the temporal evolution of individual snow parcels. Nevertheless, based on the isotopic signature observed in the layers from 1 - 4 cm and 4 - 10 cm, we assume that the winter layer is apparent at some locations during the beginning of the observation period ($\delta^{18}$O values below -40‰, Fig. 5). We can unfortunately not trace this layer at each location throughout the entire season and, hence, this dataset does not allow conclusions on whether the winter layer persists within the snowpack or diffuses with time.

The ability to combine this data set with snow depth information from DEMs enables a variety of different analyses. However, DEMs are not available for each day of the observation period (14 May to 3 August 2018) and also not for each day of snow sampling (Table 1). This complicates the quantification of the contribution of snow accumulation and erosion to the observed isotopic signal. More information on the snowfall history, especially for the time preceding the sampling, with for example observations during the winter time, and a deeper sampling might be beneficial to analyse year-round conditions of accumulation and (post-)depositional modifications of the isotope signature. Zuhr et al. (2023) compare the internal structure and the $\delta^{18}O$ composition of observed and simulated surface and sub-surface snow over the summer period in 2019 (Fig. 6 in their publication), but they also lack detailed accumulation information prior to the sampling period. The available one-point measurements of snow height evolution at the nearby AWS can provide some information on the timing of large snowfall and erosion events (Zuhr et al., 2021), but misses the spatial component.

## 4.2 Combining the stable water isotope data with other datasets

### 4.2.1 Detailed snow height information

Combining stable water isotope data with snow height information was shown to offer detailed insights in the signal formation in the upper snowpack (Zuhr et al., 2023). Detailed snow height information complementing the presented isotope dataset is available for the same site and period as the presented dataset. The snow height is discussed in Zuhr et al. (2021) and published in Zuhr et al. (2020) and Zuhr et al. (2021). The observed low $\delta^{18}$O values in the layer of 4 - 10 cm (Fig. 5) coincide with two dune features around 12 and 38 m along the transect in the beginning of the observation period. However, DEMs are not available for every day of snow sampling (Table 1), challenging a reliable assignment of individual snowfall events to specific layers throughout the season as well as individual height estimates of each sampling position. Nevertheless, the temporal sampling interval of three days or higher might reveal, together with available DEMs, insights into the near-daily evolution of the imprint and preservation of isotopic signatures and the influence of stratigraphic noise on these processes.



Data

### 4.2.2 Additional stable water isotope data from the same area

A second snow sampling scheme, referred to as surface transect, was performed in the vicinity of the photogrammetry transect during the same time in 2018 (Fig. 1b). Snow was sampled daily in three depth intervals (0 - 0.5 cm, 0 - 1 cm and 0 - 2 cm) at eleven positions with 10 m spacing along a 100 m transect. The samples were cumulatively stored in one bag per depth interval (Fig. 11). Comparing these daily sampled data with the dataset presented in this study shows, that the 3-day-sampling captures the overall trend of the isotopic composition in the surface and sub-surface snow(Fig. 11), but misses short-term fluctuations (e.g. between 30 July and 3 August). Moreover, the uppermost samples from 0 - 0.5 cm from the surface transect show more variability than the top sample from this dataset which might be caused by atmosphere-snow exchange processes or wind-driven snow redistribution. Some of the abrupt changes in $\delta^{18}O$ are associated with large latent heat fluxes (e.g. after 4 June, Fig. 3) which could indicate sublimation-driven changes in isotopes. However, we also observed snowfall events (Fig. 3) that might dominate the surface and sub-surface snow layers and their isotopic composition, especially if the amount of snowfall was larger than the sampled layer. Deciphering the individual contributions to the overall isotopic signal requires a more in-depth analysis and would benefit from more information, such as amount of snowfall and strength of snow re-distribution.

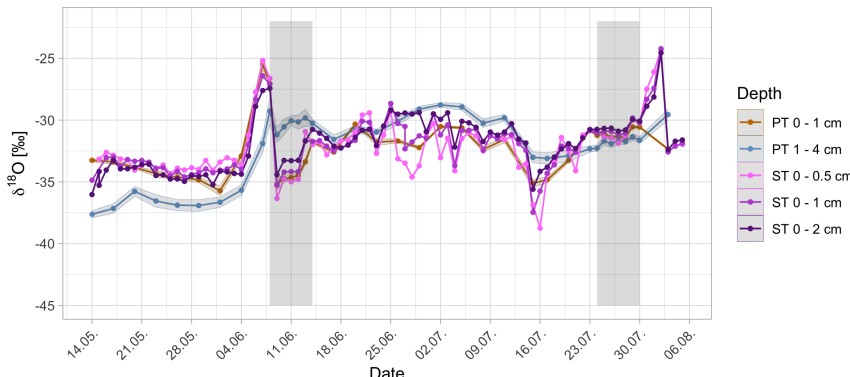

**Figure 11.** Comparison of $\delta^{18}O$ data from this study (PT; first and second layer, averaged per day and depth interval with respective standard error) and the consolidated samples from the surface transect (ST). The location of the surface transect is shown in Fig. 1b.

At the same study site, Zuhr et al. (2023) sampled the top 30 cm of the snowpack with a vertical resolution of 2 cm along a 40 m long transect (Fig. 1b and 1c in their study) between 27 May and 27 July 2019. Instead of high temporal resolution as in this study, their study sampled the upper 30 cm with higher vertical resolution, but only on six days. Their isotope dataset is accompanied by near-daily DEMs. The deeper sampling provides insights into the internal layering of the isotopic composition as well as the temporal evolution of these layers. During the 2019 season, Zuhr et al. (2023) recorded only 13 snowfall events (Fig. 2b in their study) while the season in 2018 had 29 documented snowfall events (Fig. 3). This suggests that the isotopic composition in 2018 might be more influenced by a snowfall-driven signal while their dataset from 2019 could reveal a signal formed by atmosphere-snow exchange processes, as discussed in their study. More in-depth analyses of the DEMs and other





data that indicate timing and amount of snowfall might provide a deeper understanding of the observed changes in $\delta^{18}O$ in the upper snowpack during a snowfall-dominated season.

### 4.2.3 Information on atmosphere-snow exchange processes

Besides physical modifications of the snow surface by snow erosion and redistribution, vapour-snow exchange processes con-

tribute to the surface snow isotope variability (e.g., Casado et al., 2021; Hughes et al., 2021; Wahl et al., 2022). The presented dataset is suitable to study these exchange processes by combining the detailed isotope measurements with other information and datasets. Studies with a focus on isotope and humidity fluxes have already shown that if the conditions for isotopic exchange between the atmospheric water vapour and the snow surface are given, especially in snowfall-free periods, an post-depositional atmospheric signal is introduced into the surface layer (Wahl et al., 2022). Isotope and humidity fluxes are, for

instance, available from the same study site and season (Steen-Larsen and Wahl, 2022; Steen-Larsen et al., 2022b) as well as from the nearby AWS. The large increase in temperature after 4 June is accompanied by a high LE (Fig. 2 and 6). Both could contribute to a sublimation-driven increase in $\delta^{18}O$. However, we also observed snowfall between 5 and 9 June (Fig. 2), which might dominate the isotopic composition of the surface snow and upper snowpack. Additional evidence for fractionation effects during sublimation (Madsen et al., 2019; Wahl et al., 2021) and vapour diffusion within the snowpack (snow

metamorphism) are reported in a study that analysed SSA and isotope data at the EastGRIP campsite for several years between 2016 and 2019 (Harris Stuart et al., 2023). A detailed analysis of isotopic variability from the presented dataset in conjunction with SSA changes (Steen-Larsen et al., 2022a) can help quantifying the different contributions of snowfall, redistribution, and vapour-snow exchange to the obtained isotope signal in the upper snowpack.

## 5 Conclusions

We present stable water isotope data with high temporal and spatial resolution from samples taken next to the EastGRIP campsite in northeast Greenland for the summer season in 2018. The spatio-temporal variability suggests that snow erosion and drift contribute considerably to the observed isotopic composition. The consideration of additional processes, such as vapour-snow exchange, might be necessary to capture the influence of (post-)depositional modifications on the proxy signal formation and preservation within the upper snowpack. The location of the EastGRIP campsite in the accumulation zone of

the Greenland Ice Sheet is very suitable for such investigations due to the absence of melt during the summer and a sufficiently high accumulation rate to show seasonal layering. More analyses of, for instance, combining this data with complementary datasets, e.g., snow height information and vapour-flux measurements, will contribute to an improved understanding of the climatic signal contained in stable water isotopes in firn and ice cores.



## 6 Data availability

All isotope data is available on PANGAEA via https://doi.pangaea.de/10.1594/PANGAEA.956626 (Zuhr et al., 2023). This dataset includes the variables $\delta^{18}$O, $\delta$D and d-excess as well as their respective standard deviation based on the measurement. The data are labeled with their respective location along the transect and the depth of sampling as well as the date of sampling (date/time and day-of-year).

*Author contributions.* AMZ, MH, HCSL and TL designed the study. AMZ, SW and HCSL carried out the snow sampling. AMZ performed
the measurements with the help of HM and VG. AMZ performed the analysis and prepared the manuscript with contributions from all co-authors.

*Competing interests.* The authors declare no competing interests.

*Acknowledgements.* We thank everyone who supported the field campaign at EastGRIP as well as the measurements of the stable water isotope data. This work has received funding from the European Research Council (ERC) under the European Union's Horizon 2020 re-
search and innovation program Starting Grant SPACE (Grant 716092) (recipient TL). This research has received funding from the European Research Council (ERC), European Union's Horizon 2020 research and innovation program: Starting Grant SNOWISO (Grant agreement. 759526) (recipient HCSL). EastGRIP is directed and organised by the Centre for Ice and Climate at the Niels Bohr Institute, University of Copenhagen. It is supported by funding agencies and institutions in Denmark (A. P. Møller Foundation, University of Copenhagen), USA (US National Science Foundation, Office of Polar Programs), Germany (Alfred Wegener Institute, Helmholtz Centre for Polar and
Marine Research), Japan (National Institute of Polar Research and Arctic Challenge for Sustainability), Norway (University of Bergen and Trond Mohn Foundation), Switzerland (Swiss National Science Foundation), France (French Polar Institute Paul-Emile Victor, Institute for Geosciences and Environmental research), Canada (University of Manitoba) and China (Chinese Academy of Sciences and Beijing Normal University).



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
