# Peer review of "Spatial and temporal stable water isotope data from the upper snowpack at the EastGRIP camp site, NE Greenland sampled in summer 2018"

_Earth System Science Data, 2023_

## Author Comment (AC1)

Reply to the Reviews

on the manuscript "Spatial and temporal stable water isotope data from the upper snowpack at the EastGRIP camp site, NE Greenland sampled in summer 2018" [ESSD-2023-136] by Alexandra M. Zuhr et al.

We thank both reviewers for reviewing our manuscript and providing valuable feedback on our study. We address the comments below with a point-by-point response. The original referee comments are set in normal font and our answers (author comment, AC) are set in blue.

Reply to the Review Comments of Reviewer #1

The paper presents a dataset describing the stable water isotope composition of approximately 3800 snow samples. Samples were collected at the deep drilling site of the East Greenland Ice Core Project between 14 May and 3 August 2018 along a 39 m long transect.

The dataset looks very interesting and unique in terms of detailed description, as well as for the proposed impact.

Three aspects related to the ESSD policy and requirements are major issues:

1) If this is a data description paper, ESSD states: "Although examples of data outcomes may prove necessary to demonstrate data quality, extensive interpretations of data – i.e. detailed analysis as an author might report in a research article – remain outside the scope of this data journal."

The paper matches the requested structure, but the section 2.5 introduces additional datasets complementary and outside to the presented dataset. It is ok to discuss them in the discussion section, mentioning them as references of published papers as authors have done for other datasets. Only the "high-resolution near-daily digital elevation models (DEMs) with a horizontal resolution of 1 cm and a sufficient accuracy for the purpose of this study", even published in Zuhr et al., 2021. The other data are not included in the presented dataset, available in PANGAEA.

AC: Thank you for pointing out the missing information on additional datasets. We will thoroughly read the ESSD policy again and adjust the manuscript and the presented information on additional datasets accordingly.

2) ESSD data descriptions "should instead highlight and emphasize the quality, usability, and accessibility of the dataset, database, or other data product and should describe extensive carefully prepared metadata and file structures at the data repository."

In this case I looked at data on the PANGAEA repository and the usability is limited by the position description. Authors refer to a position number, a sampling distance and a sampling

depth. They refer to a meter or submeter precision but no coordinates are available, even in the manuscript shows a coarse latitude and longitude positioning (only degree and minute for one location). The sampling depth is presented in the dataset in three different ways: as sampled depth interval, as average sample depth, and as a sample depth derived from digital elevation models. The first two parameters are redundant, thus the average depth from the surface and the sampled thickness complete the required metadata. The third parameter, expressed as percentage should be better described even in the paper.

AC: We will add the exact lat/lon position of the reference point in the manuscript. However, we note that scientifically, the exact position should not be relevant as we expect that these are general properties of the area. We will extend the description of the sample depth derived from the digital elevation model in the revised manuscript. The parameters *sampled depth interval* and *average sample depth* are not redundant, because the first is only a number (1 to 3) and describes the respective sampled layer (the first, the second or the third layer from the top) while the latter describes the sampled depth as averages, e.g. 0.005 m for the first layer sampled from 0 to 1 cm depth. We will clarify this in the data description.

3) ESSD author obligations state "Fragmentation of research papers should be avoided. A scientist who has done extensive work on a system or group of related systems should organize publication so that each paper gives a complete account of a particular aspect of the general study. It is inappropriate for an author to submit manuscripts describing essentially the same research to more than one journal of primary publication.Fragmentation of research papers should be avoided. A scientist who has done extensive work on a system or group of related systems should organize publication so that each paper gives a complete account of a particular aspect of the general study."

Authors state that "high-resolution near-daily digital elevation models (DEMs) with a horizontal resolution of 1 cm and a sufficient accuracy for the purpose of this study (RMSE of 1.3 cm (Zuhr et al., 2021))." Are the same data included in the DEM derived parameter associated with the presented dataset? The interoperability between the two datasets should be the best practice.

AC: The same DEM data, published in Zuhr et al. (2021), are used to provide absolute depth information for the isotope data presented in this manuscript. Both datasets present individual, complete and independently organized publications. However, we use the DEM data in this manuscript because they complement the isotope data. We will clarify the description and highlight the interoperability between both datasets.

Minor comments:

line 65 Whirl-Paks is a registered trademark, could it better to state the plastic bag in another way (high-purity sampling bag...?)

AC: We are mentioning WhirlPak here because we used exactly this brand and type of bag and this information might be useful for readers for reproducibility. We will extend the description by mentioning that the bags are high-purity sampling bags.

Figure 2 be consistent with Table 1: sampling modes "Normal" and "n" should be named in the same way

 AC: We will unify the description here.

Table 1 declare the year in the caption.

 AC: We will declare the year in the caption.

Figure 3 The top panel with Latent Heat Flux should have the y axis without a colour since it refers both to daily and hourly values in the plot.

AC: We will change the color scales in Figure 3.

Check references Zuhr et al 2021 and 2023 are not differentiated even they are listed both in double.

AC: We are aware of the fact that Zuhr et al 2021 and 2023 are not unique citations. Unfortunately, we were not able to solve this yet, because it seems to be an issue of LateX. With the help of the Copernicus team, we will try to solve this issue before a resubmission.

Reply to the Review Comments of Reviewer #2

The article presents an interesting study on water stable isotope variations in the upper snowpack nearby the EastGRIP drilling site in northeast Greenland. About 3800 snow samples were obtained during summer 2018 and analyzed for d18O and dD.

Other similar works were produced during last years on surface snow isotopic variability at this site.

The premise is that such a high temporal and spatial resolution in surface snow sampling will allow a better understanding of post-depositional processes.

Two high intensity period were chosen for a more frequent (daily) sampling, characterized also by a higher depth resolution of the samples, but in my opinion this, instead of providing useful data, creates a bit of confusion when compared to the rest of the dataset.

The difference in variations between different depths is higher for deuterium excess, where the upper layer shows a very different variability compared to the layers below, while for d18O the differences between those layers are not so pronounced.

The second sampling scheme, with less spatial positions but a more frequent sampling time and a higher vertical resolution, suggests that quick d-excess variations in surface snow are not captured on a 3-day sampling.

Line 2-4 I would use climatic parameters more than environmental

AC: We will change *climatic* to *environmental*.

Line 28 change "as well as the stratigraphic features at a specific site" with "as well as by the stratigraphic features at a specific site"

AC: We will change this sentence.

Line 30-34 there is a bit of confusion in this part and I suggest a rephrase: the isotopic signal is imprinted during the formation of precipitation, after snow is deposited this signal can be significantly modified by post-depositional processes occurring through exchanges between snow and atmosphere or within the snow column (isotopic diffusion)

AC: We will rephrase this part.

Line 42 change "contribute to pinning down the processes" with "has contributed in pinning down the processes"

AC: We will change this wording.

Line 43 change "and which need a better quantification" with "and need a better quantification"

AC: We will change this wording.

Line 63-64 moving the sampling position each time was certainly required in order not to resample the same point, but introduce a further variability which might not be solely due to the passing of time but also on the spatial variability given by sampling a different point

AC: We will comment on this point in the manuscript.

Line 68-70 I would rephrase as follow "The HIT 1 period, from 8 to 14 June 2018, followed a major snowfall event to study the signal imprint from the surface penetrating into the upper snowpack. During that period samples were collected only at ten sampling positions due to time constraints."

AC: We will rephrase this part.

Line 71-72 change "The second high-resolution sampling period (HIT 2) was from 24 to 30 July 2018 covering a snowfall-free period" with "The second high-resolution sampling period (HIT 2) was from 24 to 30 July 2018, covering a snowfall-free period"

AC: We will rephrase this part.

Line 73 change "with the depth intervals 0 - 0.5 cm, …" with "with the following depth intervals: 0 – 0.5 cm, …"

AC: We will change this wording.

Line 111 a root mean square deviation of 1.45‰ for d18O seems a bit too high

AC: We agree that 1.45 ‰ seems high for d18O. We assume that this might be related to the transport of the samples which were used for the intercomparison of the laboratories and/or to the different measurement setups (different vaporizer and injections into the cavity) and post-run corrections. Considering the large changes in d18O throughout the season and the differences between the depth intervals, we do not see this high RMSD as an issue.

Figure 5 It seems that snow d18O changes more in the 1-4 and 4-10 cm layers than in the upper stratum: while the first centimeter of snow is almost entirely characterized by values equal or higher than -37‰, the lower layers are affected by a higher variability; how do you explain it?
AC: We think the upper layer is largely influenced by snowfall, snowdrift and air-snow exchange processes. The deeper layers, especially the lowest layer from 4-10 cm, are characterized by snowfall events and processes during previous seasons. The values below -40 ‰, for instance, represent most likely snow from the previous winter season. With time, the depth of 4-10 cm gets mixed with snow from the spring and the summer season. Thus, the d18O values increase. It seems that the variability in this layer is larger, but it might only be the relative change in sampling depth and the cutting through different (temporal) layers.
The sampling at 12-13 m of distance starts with d18O values well below -40‰ and ends up with values above -30‰ in both 1-4 and 4-10 cm, while it shows less variation in the first layer of snow; is it possible that the snowfall event prior to the HIT 1 period changed snow d18O for the 0-1 and 1-4 layers, but it seems to affect less the 4-10 cm layer. However, I do not observe significant d18O variations following this intense snowfall. HIT 2 lack of precipitation seems to affect more the surface layer than the other two: in the first centimeter there is an overall increase in d18O in all sampling points of this period. The last days of sampling are characterized by a minor d18O decrease in all sampling points in the 0-1 cm layer, while d18O values from 1-4 and 4-10 cm show an overall increase; how do you explain it?

AC: The three layers behave indeed differently throughout the observation period. To us, it seems that, for instance, the imprint from the intense snowfall event before the HIT 1 period is first apparent in the upper most layer (0-1 cm) and is then with time transferred to the layers beneath (1-4 and 4-10 cm); however, this imprint seems to be spatially variable and not uniform across the 40 m sampling length. This might depend on the local topography because we have seen that dunes and troughs receive different amounts of snow during snowfall and snowdrift

events. It seems that in other areas, the high d18O signal is not preserved. This might be related to wind-driven redistribution and/or snowdrift after the snowfall event and/or spatial variations within the snowpack.

The difference between the upper layer (0-1 cm) and the layers beneath (1-4 and 4-10 cm) before and during the HIT 2 period might be determined by processes between the uppermost snow layer and the atmosphere due to the lack of precipitation and the possibility of sublimation and other air-snow exchange processes. The temporal sampling interval of 3 days and the vertical sampling intervals of 1 to several cm do not allow any conclusions to be drawn about small spatial scales or on the process level of sublimation/air-snow exchange.

The last sampling day shows indeed an interesting evolution of the d18O values for which we do not have a good explanation. We refrain from discussing and interpreting the data too much in the current manuscript in order to keep the publication as a data description paper.

Line 159 It is true when using a mean spatial d18O value, but when considering the single sampling points and according to the considerations I made above, I tend to disagree with this statement: at least for d18O, we see more (and faster) variability in the lower layers

AC: Considering the mean value of all 30 sampling positions for the individual layers (Fig. 6), the surface layer (0-1 cm) shows larger variations and a larger amplitude than the layers beneath. The single values for the individual layers respond differently to changes. The snowfall event in the beginning of June 2018 seems to affect the surface layer first and strongest while the layers beneath show delayed changes in their d18O values. Fig. 6 also shows the standard error of each sampling day for the respective depth layer and the values are increasing with sampling depth. We, therefore, do not fully agree that the lower layers show generally more (and faster) variability. The deeper layers show more variability over time which might be caused by the larger vertical interval covered by the samples.

Figure 6 could you use a lighter tone for temperature? It makes it difficult to correctly see the d18O dots and lines.

AC: We will change the color for temperature.

Figure 7 here it is more evident how d-excess is more sensitive to snow-atmosphere interaction and changes more frequently and more significantly in the upper layer. The period at the end of June is characterized by very d-excess values above 20, but only in the 0-1 cm stratum; these high values seem to have a low duration and decrease within few days, maybe following a snowfall event

Figure 8 for d-excess the spatial variability seems less important: the mean spatial values provide similar information to the single sampling points values presented in figure 7; this might suggest that the d-excess is more susceptible to atmospheric conditions, which are the same for the entire sampling place, than to snowfall accumulation and wind redistribution inhomogeneity

AC: We agree that the d-excess shows more variability in the upper layer than the layers beneath and also different variability than the d18O values. We refrain from providing too much

discussion and interpretation of the data because we want to keep this manuscript as a data description paper, rather than a full research article. Nevertheless, the reviewer points out interesting features in the dataset.

Figure 9 and 10 you have to specify in figure captions that you are showing mean spatial values

AC: We will add more details to the figure captions.

Line 226-228 you suggest that higher variability on the second sampling scheme is due to the atmosphere-snow exchange or wind distribution; why should that not be recorded when the snow is exposed for a longer period of time?

AC: Atmosphere-snow exchange should also be recorded in snow that is exposed for a longer period of time. However, what we want to express in this part is that a higher temporal sampling interval provides more insights into these exchange processes because of their short timescale. We expect that these exchange processes occur on a faster timescale as we have seen that, for instance, the diurnal cycle influences the isotopic composition at and close to the surface snow layer (Hughes et al., 2021). A 3-day sampling interval does not capture variations on a diurnal timescale.

References:

Hughes, A. G., Wahl, S., Jones, T. R., Zuhr, A., Hörhold, M., White, J. W. C., and Steen-Larsen, H. C.: The role of sublimation as a driver of climate signals in the water isotope content of surface snow: laboratory and field experimental results, The Cryosphere, 15, 4949–4974, https://doi.org/10.5194/tc-15-4949-2021, 2021.

Zuhr, A., Münch, T., Steen-Larsen, H. C., Hörhold, M., and Laepple, T.: Digital elevation models generated with a Structure-from-Motion photogrammetry approach at the EGRIP camp site in 2018, https://doi.org/10.1594/PANGAEA.936082, 2021.

---

## Author Response (AR1)

Reply to the Reviews

on the manuscript "Spatial and temporal stable water isotope data from the upper snowpack at the EastGRIP camp site, NE Greenland sampled in summer 2018" [ESSD-2023-136] by Alexandra M. Zuhr et al.

We thank both reviewers for reviewing our manuscript and providing valuable feedback on our study. We address the comments below with a point-by-point response. The original referee comments are set in normal font and our answers (author comment, AC) are set in blue.

Reply to the Review Comments of Reviewer #1

The paper presents a dataset describing the stable water isotope composition of approximately 3800 snow samples. Samples were collected at the deep drilling site of the East Greenland Ice Core Project between 14 May and 3 August 2018 along a 39 m long transect.

The dataset looks very interesting and unique in terms of detailed description, as well as for the proposed impact.

Three aspects related to the ESSD policy and requirements are major issues:

1) If this is a data description paper, ESSD states: "Although examples of data outcomes may prove necessary to demonstrate data quality, extensive interpretations of data – i.e. detailed analysis as an author might report in a research article – remain outside the scope of this data journal."

The paper matches the requested structure, but the section 2.5 introduces additional datasets complementary and outside to the presented dataset. It is ok to discuss them in the discussion section, mentioning them as references of published papers as authors have done for other datasets. Only the "high-resolution near-daily digital elevation models (DEMs) with a horizontal resolution of 1 cm and a sufficient accuracy for the purpose of this study", even published in Zuhr et al., 2021. The other data are not included in the presented dataset, available in PANGAEA.

AC: Thank you for pointing out the missing information on additional datasets. We adjusted the manuscript accordingly by moving the information on additional datasets to the discussion (now in section 4.2) to clearly separate the presented dataset from previously published and analyzed datasets. We discuss the interoperability of the additional datasets with the presented isotope data and mention their potential for future analyses.

2) ESSD data descriptions "should instead highlight and emphasize the quality, usability, and accessibility of the dataset, database, or other data product and should describe extensive carefully prepared metadata and file structures at the data repository."

In this case I looked at data on the PANGAEA repository and the usability is limited by the position description. Authors refer to a position number, a sampling distance and a sampling depth. They refer to a meter or submeter precision but no coordinates are available, even in the manuscript shows a coarse latitude and longitude positioning (only degree and minute for one location). The sampling depth is presented in the dataset in three different ways: as sampled depth interval, as average sample depth, and as a sample depth derived from digital elevation models. The first two parameters are redundant, thus the average depth from the surface and the sampled thickness complete the required metadata. The third parameter, expressed as percentage should be better described even in the paper.

AC: We added the exact position of one corner of the transect to the manuscript and the dataset in PANGAEA. However, we note that scientifically, the exact position should not be relevant as we expect that these are general properties of the area. Additionally, we extended the description of the parameters in the dataset archived in PANGAEA. We moved the section describing the dataset to the Results section in the manuscript (now section 3.1 Description of the data). It reads the following:

> The dataset is archived in PANGAEA (Zuhr et al., 2023a) and contains columns indicating the sampled depth interval (*Depth layer ice/snow* with the values 1, 2 and 3 indicating the layers 0 - 1 cm, 1 - 4 cm and 4 - 10 cm), the average sample depth (*Depth ice/snow [m]* with the values 0.005 m, 0.025 m and 0.07 m for the three respective layers) and the relative depth derived from additional information on the surface height (*Depth rel.* in meters with data from Zuhr et al. (2021a)). The sampling position along the transect is indicated (*Position* with numbers from 1 to 30) as well as the absolute distance along the transect (*Dist [m]*).

3) ESSD author obligations state "Fragmentation of research papers should be avoided. A scientist who has done extensive work on a system or group of related systems should organize publication so that each paper gives a complete account of a particular aspect of the general study. It is inappropriate for an author to submit manuscripts describing essentially the same research to more than one journal of primary publication.Fragmentation of research papers should be avoided. A scientist who has done extensive work on a system or group of related systems should organize publication so that each paper gives a complete account of a particular aspect of the general study."

Authors state that "high-resolution near-daily digital elevation models (DEMs) with a horizontal resolution of 1 cm and a sufficient accuracy for the purpose of this study (RMSE of 1.3 cm (Zuhr et al., 2021))." Are the same data included in the DEM derived parameter associated with the presented dataset? The interoperability between the two datasets should be the best practice.

AC: The same DEM data, published in Zuhr et al. (2021a), are used to provide absolute depth information for the isotope data presented in this manuscript. Both datasets present individual, complete and independently organized publications. However, we use the DEM data in this manuscript because they complement the isotope data by providing information on the evolution of the snow height. The entire paragraph is moved to the discussion and modified now in

section 4.2. Moreover, we extended the description and highlighted the interoperability between both datasets.

Minor comments:

line 65 Whirl-Paks is a registered trademark, could it better to state the plastic bag in another way (high-purity sampling bag...?)

AC: We are mentioning WhirlPak here because we used exactly this brand and type of bag and this information might be useful for readers for reproducibility. We extended the description by mentioning that the bags are high-purity sampling bags.

Figure 2 be consistent with Table 1: sampling modes "Normal" and "n" should be named in the same way

 AC: We adapted the description.

Table 1 declare the year in the caption.

 AC: We are now mentioning the year in the caption.

Figure 3 The top panel with Latent Heat Flux should have the y axis without a colour since it refers both to daily and hourly values in the plot.

AC: We changed the color of the y-axis for Latent Heat Flux in Figure 3.

Check references Zuhr et al 2021 and 2023 are not differentiated even they are listed both in double.

AC: We fixed the references. It now reads Zuhr et al. (2021a) and Zuhr et al. (2021b) as well as Zuhr et al. (2023a) and Zuhr et al. (2023b).

The article presents an interesting study on water stable isotope variations in the upper snowpack nearby the EastGRIP drilling site in northeast Greenland. About 3800 snow samples were obtained during summer 2018 and analyzed for d18O and dD.

Other similar works were produced during last years on surface snow isotopic variability at this site.

The premise is that such a high temporal and spatial resolution in surface snow sampling will allow a better understanding of post-depositional processes.

Two high intensity period were chosen for a more frequent (daily) sampling, characterized also by a higher depth resolution of the samples, but in my opinion this, instead of providing useful data, creates a bit of confusion when compared to the rest of the dataset.

The difference in variations between different depths is higher for deuterium excess, where the upper layer shows a very different variability compared to the layers below, while for d18O the differences between those layers are not so pronounced.

The second sampling scheme, with less spatial positions but a more frequent sampling time and a higher vertical resolution, suggests that quick d-excess variations in surface snow are not captured on a 3-day sampling.

Line 2-4 I would use climatic parameters more than environmental

AC: We changed *climatic* to *environmental*.

Line 28 change "as well as the stratigraphic features at a specific site" with "as well as by the stratigraphic features at a specific site"

AC: We changed this sentence.

Line 30-34 there is a bit of confusion in this part and I suggest a rephrase: the isotopic signal is imprinted during the formation of precipitation, after snow is deposited this signal can be significantly modified by post-depositional processes occurring through exchanges between snow and atmosphere or within the snow column (isotopic diffusion)

AC: We rephrased this part with the suggestion.

Line 42 change "contribute to pinning down the processes" with "has contributed in pinning down the processes"
Line 43 change "and which need a better quantification" with "and need a better quantification"

AC: We changed the wording for the entire sentence with the suggested phrases. It reads now "These and many other studies have contributed in pinning down the processes which are not fully understood yet and need a better quantification."

Line 63-64 moving the sampling position each time was certainly required in order not to resample the same point, but introduce a further variability which might not be solely due to the passing of time but also on the spatial variability given by sampling a different point

AC: We are pointing to this uncertainty in section 4.1 Limitations of the dataset now with the following sentence:

> Moreover, spatial variability introduced by moving the sampling position each time might cause additional uncertainty which should be considered when the aim is to analyse (near-)daily changes in the isotopic composition from nearby sampling locations.

Line 68-70 I would rephrase as follow "The HIT 1 period, from 8 to 14 June 2018, followed a major snowfall event to study the signal imprint from the surface penetrating into the upper snowpack. During that period samples were collected only at ten sampling positions due to time constraints."
Line 71-72 change "The second high-resolution sampling period (HIT 2) was from 24 to 30 July 2018 covering a snowfall-free period" with "The second high-resolution sampling period (HIT 2) was from 24 to 30 July 2018, covering a snowfall-free period"
Line 73 change "with the depth intervals 0 - 0.5 cm, …" with "with the following depth intervals: 0 – 0.5 cm, …"

AC: We rephrased both parts with the suggested wording.

Line 111 a root mean square deviation of 1.45‰ for d18O seems a bit too high

AC: We agree that 1.45 ‰ seems high for $d^{18}O$. We assume that this might be related to the transport of the samples which were used for the intercomparison of the laboratories and/or to the different measurement setups (different vaporizer and injections into the cavity) and post-run corrections. Considering the large changes in $d^{18}O$ throughout the season and the differences between the depth intervals, we do not see this high RMSD as an issue.

Figure 5 It seems that snow d18O changes more in the 1-4 and 4-10 cm layers than in the upper stratum: while the first centimeter of snow is almost entirely characterized by values equal or higher than -37‰, the lower layers are affected by a higher variability; how do you explain it?
AC: We think the upper layer is largely influenced by snowfall, snowdrift and air-snow exchange processes while the deeper layers, especially the lowest layer from 4-10 cm, are characterized by snowfall events and processes during previous seasons. Thus, we see different processes in the individual depth intervals. Indeed, it seems that the variability in the lowest layer is larger, which might be due to changes in the relative sampling depth and the cutting through different (temporal) layers.
We extended Figure 5 and 7 with an information on the snow height evolution throughout the season in 2018. The snow height indicates that the changes at the surface might be driven by new snow accumulation while the changes in the lowest sampled layer might be caused by sampling different internal layers. We refrain from extending the discussion at this point to keep the manuscript as a data (description) publication and not a research article.

The sampling at 12-13 m of distance starts with d18O values well below -40‰ and ends up with values above -30‰ in both 1-4 and 4-10 cm, while it shows less variation in the first layer of snow; is it possible that the snowfall event prior to the HIT 1 period changed snow d18O for the 0-1 and 1-4 layers, but it seems to affect less the 4-10 cm layer. However, I do not observe significant d18O variations following this intense snowfall. HIT 2 lack of precipitation seems to affect more the surface layer than the other two: in the first centimeter there is an overall increase in d18O in all sampling points of this period. The last days of sampling are characterized by a minor d18O decrease in all sampling points in the 0-1 cm layer, while d18O values from 1-4 and 4-10 cm show an overall increase; how do you explain it?

AC: The three layers behave indeed differently throughout the observation period. To us, it seems that, for instance, the imprint from the intense snowfall event before the HIT 1 period is first apparent in the upper most layer (0-1 cm) and is then with time transferred to the layers beneath (1-4 and 4-10 cm); however, this imprint seems to be spatially variable and not uniform across the 40 m sampling length.
We mention in the manuscript (section 4.2 in the new version) that several processes can be responsible for changes in the stable water isotopic composition: metamorphism (Casado et al., 2021; Harris Stuart et al., 2023); wind-driven redistribution and variable snow deposition (Zuhr et al., 2021b and 2023b); snow-air exchange processes (Hughes et al., 2021; Wahl et al., 2022).

Line 159 It is true when using a mean spatial d18O value, but when considering the single sampling points and according to the considerations I made above, I tend to disagree with this statement: at least for d18O, we see more (and faster) variability in the lower layers

AC: Considering the mean value of all 30 sampling positions for the individual layers (Fig. 6), the surface layer (0-1 cm) shows larger variations and a larger amplitude than the layers beneath. However, the standard deviation and the variance of each layer increases with depth for $d^{18}O$ indicating larger variations in deeper layers while single values from individual layers may respond differently to changes due to different reasons. The deepest layer does not only show variations over time for the same depth, but also the vertical change over time due to new snow accumulation from the top and the cutting through different layers.
We adjusted the text in the manuscript accordingly and are now mentioning values for standard deviation and variance for each sampled depth layer.

Figure 6 could you use a lighter tone for temperature? It makes it difficult to correctly see the d18O dots and lines.

AC: We changed the color for temperature in Figures 6 and 8.

Figure 7 here it is more evident how d-excess is more sensitive to snow-atmosphere interaction and changes more frequently and more significantly in the upper layer. The period at the end of June is characterized by very d-excess values above 20, but only in the 0-1 cm stratum; these high values seem to have a low duration and decrease within few days, maybe following a snowfall event

Figure 8 for d-excess the spatial variability seems less important: the mean spatial values provide similar information to the single sampling points values presented in figure 7; this might suggest that the d-excess is more susceptible to atmospheric conditions, which are the same for the entire sampling place, than to snowfall accumulation and wind redistribution inhomogeneity

AC: We agree that the d-excess shows more variability in the upper layer than the layers beneath and also different variability than the d$^{18}$O values. The reviewer points out interesting features in the dataset, but we want to keep this manuscript as a data description paper and to not provide more discussion at this point.

Figure 9 and 10 you have to specify in figure captions that you are showing mean spatial values

AC: We modified the caption and are mentioning now that the data are spatial averages.

Line 226-228 you suggest that higher variability on the second sampling scheme is due to the atmosphere-snow exchange or wind distribution; why should that not be recorded when the snow is exposed for a longer period of time?

AC: Atmosphere-snow exchange should also be recorded in snow that is exposed for a longer period of time. The two sampling schemes that we compare here have a different depth resolution for the layers close to the surface. Hence, we expect that snow-atmosphere surface exchange signals will be most visible in the thin 0-0.5cm surface layer from the surface transect and already more damped in the 0-1 cm layer from the normal sampling scheme in this study. Additionally, we want to express in this part that a higher temporal sampling interval provides more insights into these exchange processes because of their shorter timescale. We expect that these exchange processes occur on faster timescales as we have seen that, for instance, the diurnal cycle influences the isotopic composition at and close to the surface snow layer (Hughes et al., 2021; Wahl et al., 2022). Our 3-day sampling interval might not capture these fast variations.

**References**

Casado, M., Landais, A., Picard, G., Arnaud, L., Dreossi, G., Stenni, B., and Prié, F.: Water Isotopic Signature of Surface Snow Metamorphism in Antarctica, Geophysical Research Letters, 48, e2021GL093 382, https://doi.org/10.1029/2021GL093382, 2021.

Harris Stuart, R., Faber, A.-K., Wahl, S., Hörhold, M., Kipfstuhl, S., Vasskog, K., Behrens, M., Zuhr, A. M., and Steen-Larsen, H. C.: Exploring the role of snow metamorphism on the isotopic composition of the surface snow at EastGRIP, The Cryosphere, 17, 1185–1204, https://doi.org/10.5194/tc-17-1185-2023, 2023

Hughes, A. G., Wahl, S., Jones, T. R., Zuhr, A., Hörhold, M., White, J. W. C., and Steen-Larsen, H. C.: The role of sublimation as a driver of climate signals in the water isotope content of surface snow: laboratory and field experimental results, The Cryosphere, 15, 4949–4974, https://doi.org/10.5194/tc-15-4949-2021, 2021.

Wahl, S., Steen-Larsen, H. C., Hughes, A. G., Dietrich, L. J., Zuhr, A., Behrens, M., Faber, A.-K., and Hörhold, M.: Atmosphere-Snow Exchange Explains Surface Snow Isotope Variability, Geophysical Research Letters, 49, https://doi.org/https://doi.org/10.1029/2022GL099529, 2022.

Zuhr, A. M., Münch, T., Steen-Larsen, H. C., Hörhold, M., and Laepple, T.: Digital elevation models generated with a Structure-from-Motion photogrammetry approach at the EGRIP camp site in 2018, PANGAEA, https://doi.org/10.1594/PANGAEA.936082, 2021a.

Zuhr, A. M., Münch, T., Steen-Larsen, H. C., Hörhold, M., and Laepple, T.: Local-scale deposition of surface snow on the Greenland ice sheet, The Cryosphere, 15, 4873–4900, https://doi.org/10.5194/tc-15-4873-2021, 2021b.

Zuhr, A. M., Wahl, S., Steen-Larsen, H. C., Faber, A.-K., Behrens, M., Zolles, T., Meyer, H., Gkinis, V., Weiner, M., Sporr, S., and Laepple, T.: Stable water isotopes in snow from a regular sampling of the upper 10 cm at the EastGRIP deep drilling site during the 2018 summer season, PANGAEA, https://doi.org/10.1594/PANGAEA.956626, 2023a.

Zuhr, A. M., Wahl, S., Steen-Larsen, H. C., Hörhold, M., Meyer, H., and Laepple, T.: A Snapshot on the Buildup of the Stable Water Isotopic Signal in the Upper Snowpack at EastGRIP on the Greenland Ice Sheet, Journal of Geophysical Research: Earth Surface, 128, 410 e2022JF006, https://doi.org/https://doi.org/10.1029/2022JF006767, 2023b.